# Optical Properties and Concentration Quenching Mechanism of Er^3+^ Heavy Doped Gd_2_(MoO_4_)_3_ Phosphor for Green Light-Emitting Diode

**DOI:** 10.3390/nano12203641

**Published:** 2022-10-17

**Authors:** Dongyu Li, Bing Xu, Zhen Huang, Xiao Jin, Zhenghe Zhang, Tingting Zhang, Deng Wang, Xuping Liu, Qinghua Li

**Affiliations:** 1Guangdong Provincial Key Laboratory of Development and Education for Special Needs Children, Zhanjiang 524048, China; 2School of Physics Science and Technology, Lingnan Normal University, Zhanjiang 524048, China; 3Institute of High Energy Physics and National Center for Nanoscience and Technology, Chinese Academy of Sciences, Beijing 100049, China

**Keywords:** heavy doped, upconversion, concentration quenching, Er^3+^ concentration

## Abstract

Upconversion materials capable of converting low-energy excitation photons into high-energy emission photons have attracted considerable interest in recent years. However, the low upconversion luminescence seriously hinders the application of upconversion phosphors. Heavy lanthanide doping without concentration quenching represents a direct and effective method to enhance the emission intensity. In this study, Er^3+^ heavy doped Gd_2_(MoO_4_)_3_ phosphor with a monoclinic phase was prepared by a sol–gel process. Under excitation at 976 nm, Gd_2_(MoO_4_)_3_:Er^3+^ phosphor emitted remarkably intense green emission, and Er^3+^ concentration up to 20 mol% did not cause concentration quenching. Here, we discuss the upconversion mechanism and concentration quenching. When the Er^3+^ concentration was in the range of 30–60 mol%, the concentration quenching was governed by the electric dipole–dipole interaction, and when the concentration was greater than 60 mol%, the concentration quenching was controlled by the exchange interactions. The result provides a schematic basis for identifying a phosphor host with heavy lanthanide doping.

## 1. Introduction

Lanthanide-doped upconversion materials capable of converting low-energy near-infrared light to high-energy visible light are potential candidates for white light displays [1], upconversion lasers [2], solar cells [3], anticounterfeit labels [4], photocatalytic structures [5], volumetric color imaging [6], temperature measurement [7], super-resolution imaging [8], and biological fluorescence labeling [9]. However, some problems limit their applications, including low upconversion emission intensity.

Enhancing upconversion emission intensity by heavy doping is a straightforward method [10]. High concentrations of sensitizer and activator ions contribute to improved absorption of excitation light and emission intensity. Generally, the doping concentration of the activator should not exceed 2 mol%; otherwise, it will cause concentration quenching and reduce the emission intensity [11,12]. Concentration quenching can be eliminated by employing core-shell structures in sensitizer Yb^3+^/RE^3+^ (RE = Er, Tm, Ho activators) systems [10,13,14,15,16]. However, this approach to enhancing upconversion emission intensity is limited, as the host lattice cannot easily accommodate a large number of dopant ions. Therefore, it is imperative to develop host materials that allow for heavy lanthanide doping.

Gd_2_(MoO_4_)_3_ is an excellent host due to its relatively low maximum phonon energy and good thermal stability. In addition, Gd^3+^ ions and activated ions (RE^3+^) are both rare earth ions with similar chemical properties and can achieve high-concentration doping in a Gd_2_(MoO_4_)_3_ host, increasing the number of luminescent centers and enhancing the intensity of upconversion luminescence. Few reports have been published to date on the upconversion luminescence of Er^3+^ single-doped Gd_2_(MoO_4_)_3_ [17]. The upconversion luminescence properties of Gd_2_(MoO_4_)_3_:Er^3+^ phosphors had been investigated under excitation in the range of 1510–1565 nm [17]. However, for the most widely used ~980 nm excitation wavelength, it is necessary to study the upconversion luminescence characteristics and conduct concentration quenching analysis of Er^3+^ heavy doped Gd_2_(MoO_4_)_3_ phosphors.

In this study, Er^3+^ heavy doped Gd_2_(MoO_4_)_3_ phosphor with a monoclinic phase was prepared by a sol–gel method. The upconversion mechanism of Gd_2_(MoO_4_)_3_:Er^3+^ phosphor was studied under the excitation of a 976 nm diode laser (LD). Furthermore, the concentration quenching mechanism was analyzed.

## 2. Materials and Methods

Gd_2_(MoO_4_)_3_:*x*Er^3+^ (*x* = 1, 5, 10, 15, 20, 25, 30, 35, 40, 50, 60, 70, 80, 90 mol%) phosphors were prepared using a simple sol–gel process. All raw materials were purchased from Sinopharm. According to our previous work [18], Gd(NO_3_)_3_·6H_2_O (AR), (NH_4_)_6_Mo_7_O_24_ (AR), and Er(NO_3_)_3_·6H_2_O (AR) with precalculated concentrations were dissolved in deionized water, and citric acid (AR) was added to this solution. Subsequently, the pH level of the solution was adjusted to about 7 by adding an appropriate amount of ammonia (AR). The resulting solution was heated at 80 °C under continuous magnetic vigorous stirring for 0.5 h and dried in a drying box at 130 °C for 20 h. Finally, the dried gel was placed in a muffle furnace at 800 °C for 2 h.

The crystalline phases were identified by an X-ray diffractometer (Empyrean, Panalytical, The Netherlands) with Cu Kα radiation (λ = 1.5418 Å) at a scanning rate of 4° per minute. A scanning electron microscope (SEM) (JSM-5510; JEOL, Tokyo, Japan) equipped with an energy-dispersive X-ray spectrum (EDS) analyzer was used to characterize the microstructures. The upconversion emission spectra were measured using an *i*HR550 spectrofluorometer.

## 3. Results and Discussion

### 3.1. Crystal Structure and Morphology

The XRD patterns of Gd_2_(MoO_4_)_3_:*x*Er^3+^ phosphors are shown in Figure 1a. The XRD patterns are well-matched with the standard data (JCPDS No. 26-0655), indicating that a single monoclinic phase, Gd_2_(MoO_4_)_3_:Er^3+^ phosphor, is acquired [19]. The monoclinic Gd_2_(MoO_4_)_3_ has a space group of C2/c with the following cell parameters: *a* = 7.53, *b* = 11.38, *c* = 11.40 Å, and β = 109.32 degrees. The stronger diffraction peaks are located at 28.42° and 29.4°, corresponding to the (–221) and (023) crystal planes, respectively. No other impurity peaks are observed, indicating that the doped Er^3+^ ions have been incorporated into the lattice, replacing the lattice position of Gd^3+^. The morphology and the elemental components of Gd_2_(MoO_4_)_3_:20Er^3+^ phosphor were identified by SEM and EDS measurements, as shown in Figure 1b,c. The prepared phosphor is composed of tightly packed particles with relatively uniform morphology, and the average particle size is about 0.9 μm (see inset of Figure 1b). Such morphology is commonly observed in molybdate and tungstate phosphors synthesized by the sol–gel method with subsequent annealing [20,21,22]. The EDS result confirms that the elemental components are Gd, Mo, Er, and O.

### 3.2. Upconversion Mechanism Analysis

The upconversion emission properties of Gd_2_(MoO_4_)_3_:Er^3+^ phosphor depend strongly on the Er^3+^ doping concentration. To reveal the concentration-dependent upconversion luminescence properties, the upconversion luminescence spectra of Gd_2_(MoO_4_)_3_:*x*Er^3+^ phosphors under 976 nm LD excitation are shown in Figure 2a. When the Er^3+^ concentration increases to 20 mol%, the green and red upconversion emission intensities are the strongest, decreasing with increasing Er^3+^ concentration. The intense green and weak red emissions centered at 531, 552, and 668 nm are attributed to the ^2^H_11/2_→ ^4^I_15/2_, ^4^S_3/2_ → ^4^I_15/2_, and ^4^F_9/2_ → ^4^I_15/2_ transitions of Er^3+^, respectively. The green emission intensity is quite strong in comparison to the red emission intensity, as shown in Figure 2b. Compared with the Gd_2_(MoO_4_)_3_:1Er^3+^ phosphor, the green and red emission intensities of Gd_2_(MoO_4_)_3_:20Er^3+^ phosphor increase by about sixfold and twofold, respectively (see Figure 2c). As the Er^3+^ concentration increases from 20 to 60 mol%, the luminescence intensity decreases rapidly due to concentration quenching. When the concentration of Er^3+^ increases from 60 to 90 mol%, the green emission intensity decreases slowly as a result of the concentration quenching effect. The red emission intensity increases in this concentration range, possibly because at such a high Er^3+^ ion concentration, the distance between Er^3+^ ions is reduced, and the interaction is enhanced. Through cross relaxation, ^2^H_11/2_ → ^4^I_9/2_ and ^4^I_15/2_ → ^4^I_13/2_, the layout of the intermediate energy level (^4^I_13/2_) results in the enhancement of red emission [23]. When the Er^3+^ concentration in the Gd_2_(MoO_4_)_3_ host is as high as 20 mol%, concentration quenching has not occurred, indicating that Gd_2_(MoO_4_)_3_ is an excellent host material. This may be because the chemical properties of Er^3+^ ions and those of Gd^3+^ ions are similar, and their radii are almost the same. Therefore, Er^3+^ ions can easily replace Gd^3+^ ions in the matrix and enter the lattice.

Figure 3a shows the intensity ratio of green and red light (GRR) in Gd_2_(MoO_4_)_3_:*x*Er^3+^ phosphors. With increased Er^3+^ concentration from 1 to 20 mol%, the GRR correspondingly increases from 33.2 to 105.2. However, as the Er^3+^ concentration continues to increase, GRR shows a decreasing trend. When the concentration of Er^3+^ is 90 mol%, the GRR is 42.4. Due to its high GRR, Er^3+^-doped Gd_2_(MoO_4_)_3_ powder is a high-quality green phosphor. The calculated chromaticity coordinate of Gd_2_(MoO_4_)_3_: 20Er^3+^ phosphor is X = 0.23 and Y = 0.74, falling exactly in the green region of the CIE chromaticity diagram (see Figure 3b).

To understand the upconversion processes in Gd_2_(MoO_4_)_3_:Er^3+^ phosphor, the excitation power dependence of the green and red emission intensities was measured (see Figure 4a). In the upconversion process, the upconversion emission intensity is proportional to the n value of pumping laser power [24]
(1)$$I\propto P^{n}$$
where *I* is the upconversion emission intensity, *P* is the pump laser power, and n is the number of pumping photons required in the upconversion process. The slopes (*n* value) are 1.8 and 1.3 for the green and red upconversion emissions, respectively. The n values indicate that two photons are involved in the green and red upconversion emissions. The possible upconversion mechanism for Gd_2_(MoO_4_)_3_:Er^3+^ phosphor is shown in Figure 4b, and the detailed upconversion mechanism was discussed in our previous work [18]. The ^2^H_11/2_/^4^S_3/2_ and ^4^F_9/2_ levels of Er^3+^ ions are populated via multiphonon relaxation from the ^4^F_7/2_ level, and the green and red upconversion emissions are observed. There are two possible upconversion pathways for population of the ^4^F_7/2_ level of Er^3+^. The first is the electronic transition from ground state ^4^I_15/2_ to the ^4^I_11/2_ level via ground-state absorption (GSA1), with further excitation to the ^4^F_7/2_ level via excited-state absorption (ESA1). The second possible upconversion pathway is a high excited state energy transfer (HESET). The Er^3+^–MoO_4_^2−^ dimer complex absorbs an infrared photon and is excited from the ground state |^4^I_15/2_, ^1^A_1_> to the intermediate state |^4^I_11/2_, ^1^A_1_> via GSA2 under 976 nm excitation. Subsequently, the |^4^I_15/2_, ^1^T_1_> state is further excited from the |^4^I_11/2_, ^1^A_1_> state via ESA3 and then decays nonradiatively to the |^4^I_15/2_, ^3^T_2_> state. The energy transfer from |^4^I_15/2_, ^3^T_2_> to the ^4^F_7/2_ level of Er^3+^ via the HESET process is efficient. Thus, the populated ^4^F_7/2_ level relaxes nonradiatively by a fast multiphonon decay process to the ^2^H_11/2_/^4^S_3/2_ and ^4^F_9/2_ levels. Then, the electrons transition from the excited state (^2^H_11/2_/^4^S_3/2_ and ^4^F_9/2_) to the ground state, accompanied by intense green and weak red emissions, in agreement with our experimental results (see Figure 2a). Furthermore, the ^4^F_9/2_ (Er^3+^) is populated via ESA2, which involves the ^4^F_9/2_ ← ^4^I_13/2_ transition.

### 3.3. Concentration Quenching Analysis

As previously stated, Gd_2_(MoO_4_)_3_:Er^3+^ phosphor emitted remarkably intense green emission under excitation at 976 nm, and Er^3+^ concentration up to 20 mol% did not cause concentration quenching. Typically, the optimal doping concentration of Er^3+^ ions does not exceed 2 mol% to reduce excitation energy loss due to cross relaxation. Why is the Er^3+^ doping concentration of Gd_2_(MoO_4_)_3_ so high? According to the Dexter theory, the functional relationship between the fluorescence intensity (*I*) and the activator concentration (*x*) can be expressed as [25,26,27]:
(2)$$I\propto \left(1+A\right)/\gamma \left[\alpha^{1-s}/3\Gamma \left(1+s/3\right)\right]\left(\alpha \geq 1\right)$$
where $\alpha =x{\left[\left(1+A\right)X_{0}/\gamma \right]}^{3/s}\Gamma \left(1-s/3\right)\propto x$; Γ(1 − s/3) is a Γ function; simplies the interaction mechanism between Er^3+^ ions; *s* = 3, 6, 8, and 10 denote exchange interaction, electric dipole-dipole interaction, electric dipole-quadrupole interaction, and electric quadrupole-quadrupole interaction, respectively; *γ* is the intrinsic transition probability of the donor; *I* is the upconversion emission intensity for ^2^H_11/2_/^4^S_3/2_ → ^4^I_15/2_ transitions of Er^3+^; *x* is the Er^3+^ concentration; and A and X_0_ are constants. Given the logarithm operation on both sides of Equation (2), the following can be derived [27,28]:(3)$$\mathrm{lg}\left(I/x\right)=-\left(s/3\right)\cdot \mathrm{lg}x+b$$
where *b* is the constant. Plots of lg(*I*/*x*) versus lg*x* are depicted in Figure 5 based on the upconversion emission spectra. The concentration quenching phenomenon occurs only when the Er^3+^ concentration is greater than 20 mol%, so the selected concentration is in the range of 30 to 90 mol%. When the Er^3+^ concentration is 30–60 mol%, the slope obtained by the fitting is −1.96, and the corresponding *s* value is 5.88 (1.96 × 3), which is very close to the theoretical value of 6 for the electric dipole–dipole interaction (see Figure 5). Therefore, in Gd_2_(MoO_4_)_3_:Er^3+^ phosphor, the fluorescence quenching caused by Er^3+^ concentration in the range of 30–60 mol% is governed by the electric dipole–dipole interaction. When the Er^3+^ concentration is increased from 60 to 90 mol%, the corresponding s value is 4.32 (1.44 × 3), which approaches the theoretical value of 3 for the exchange interactions. Therefore, it is inferred that in this concentration range, the fluorescence quenching is mainly caused by the exchange interaction.

To obtain insight into the concentration quenching, Dexter also pointed out that the type of activator ion interaction is crucially determined by the distance between the activator ions [25]. Blasse and Grabmaier calculated that when the critical distance (*R*c) between the activator ions is in the range of 5–8 Å, the fluorescence quenching is caused by the exchange interaction, and when the *R*_c_ is about 3 nm, the fluorescence quenching is caused by the electric dipole–dipole interaction. *R*_C_ can be calculated according to the following formula [29]:(4)$$R_{C}\approx 2{\left(\frac{3V}{4\pi x_{C}N}\right)}^{1/3}$$
where *V* is the unit cell volume, *x*_C_ is the critical concentration of activator ions, and *N* refers to the number of cations in the unit cell. The monoclinic Gd_2_(MoO_4_)_3_ with space group C2/c (*a* = 7.55, *b* = 11.44, *c* = 11.47 Å, *V* = 934.51 Å^3^, β = 109.32 degree, *Z* = 4) is constructed by GdO_8_ and MoO_4_ polyhedral groups. According to the above inference, the Er^3+^ ion concentration of 60 mol% is the turning point of the Er^3+^ ion exchange interaction and electric dipole–dipole interaction. Substituting *x*_C_ = 0.6 into equation (4) yields *R*_C_ = 9 Å. This distance is very close to the critical distance of 8 Å for the exchange interaction. The radius of Er^3+^ ions is 0.89 Å, which is slightly smaller than the Gd^3+^ ion radius of 0.94 Å. When the Er^3+^ concentration is as high as 60 mol%, it is reasonable to believe that the calculated *R*_C_ is larger than the real value. That is, when the Er^3+^ ion concentration is 60 mol%, the exchange interaction has already occurred. This further verifies the correctness of the above inferences. Due to the unique lattice structure of Gd_2_(MoO_4_)_3_, the distance between the activated ions is large, allowing for higher concentrations to be incorporated without concentration quenching. Combined with the model analysis, this result can also provide some theoretical guidance to identify a host that allows for the incorporation of high concentrations of rare earth elements.

## 4. Conclusions

In summary, Gd_2_(MoO_4_)_3_:*x*Er^3+^ phosphors with a monoclinic phase were prepared by a simple sol–gel method. The phosphors exhibit intense upconversion green emission under 976 nm excitation. The delayed quenching concentration for ^2^H_11/2_/^4^S_3/2_ → ^4^I_15/2_ transition of Er^3+^ reaches up to 20 mol%. When the Er^3+^ concentration is in the range of 30–60 mol%, the concentration quenching progress is governed by the electric dipole–dipole interaction, and when the Er^3+^ concentration is greater than 60 mol%, concentration quenching is governed by the exchange interaction. The results show that Gd_2_(MoO_4_)_3_:Er^3+^ powder is an excellent green phosphor, and Gd_2_(MoO_4_)_3_ material is an excellent host for lanthanide heavy doping.

## Figures and Tables

**Figure 1 nanomaterials-12-03641-f001:**
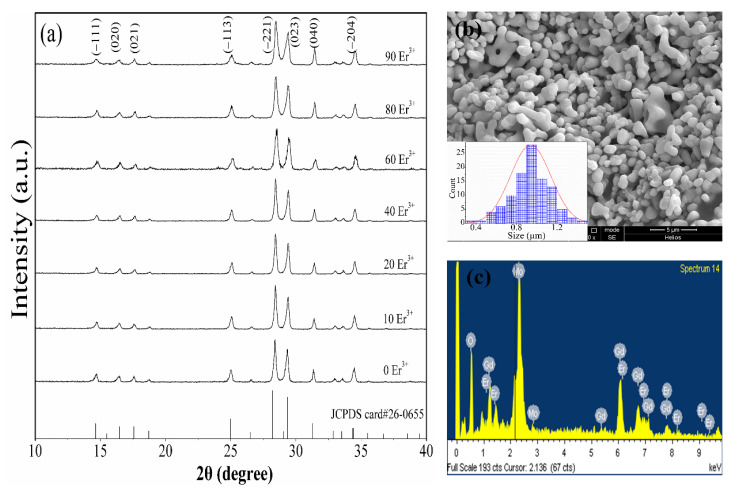
XRD patterns of Gd_2_(MoO_4_)_3_:*x*Er^3+^ (*x* = 0, 10, 20, 40, 60, 80, 90 mol%) phosphors (**a**). SEM image (**b**) and EDS spectrum (**c**) of Gd_2_(MoO_4_)_3_:20Er^3+^ phosphor. The inset of (**b**) shows the particle size distribution.

**Figure 2 nanomaterials-12-03641-f002:**
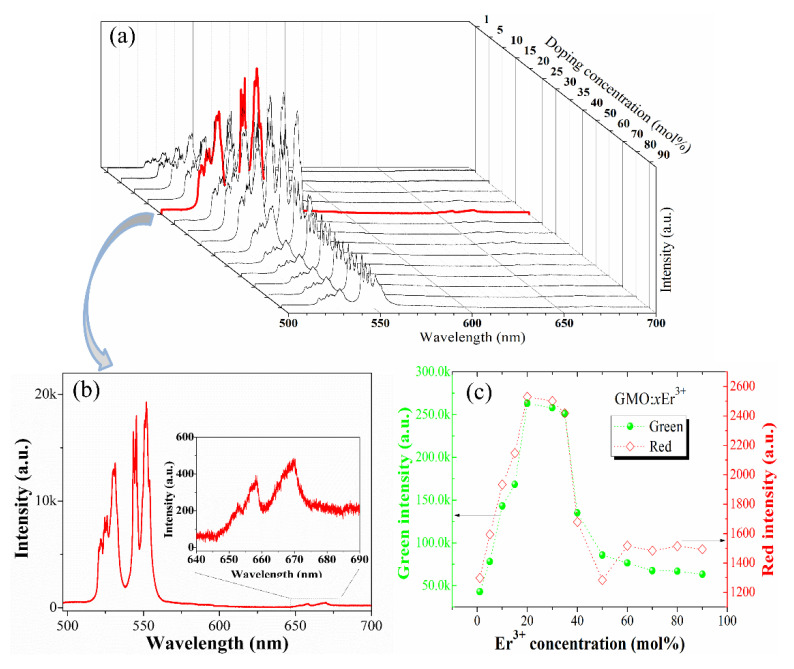
Upconversion emission spectra of Gd_2_(MoO_4_)_3_:*x*Er^3+^ phosphors (**a**). Upconversion emission spectrum of Gd_2_(MoO_4_)_3_:20Er^3+^ phosphor; inset: the red emission band (**b**). The intensities of green emission (left) and red emission (right) as a function of Er^3+^ concentration in Gd_2_(MoO_4_)_3_:*x*Er^3+^ phosphors (**c**).

**Figure 3 nanomaterials-12-03641-f003:**
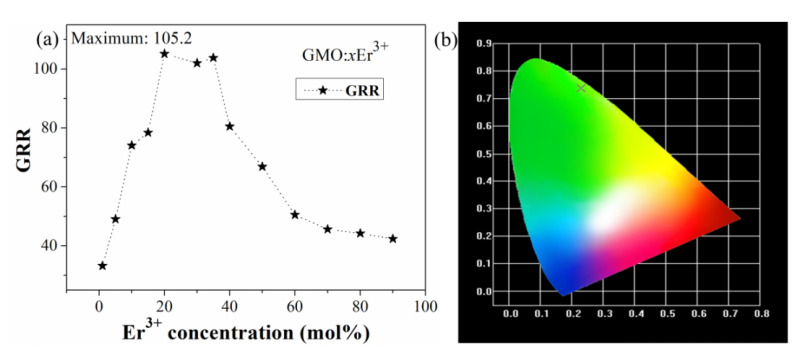
GRR as a function of Er^3+^ concentration in Gd_2_(MoO_4_)_3_:*x*Er^3+^ phosphors (**a**). CIE chromaticity diagram of Gd_2_(MoO_4_)_3_:20Er^3+^ phosphor (**b**).

**Figure 4 nanomaterials-12-03641-f004:**
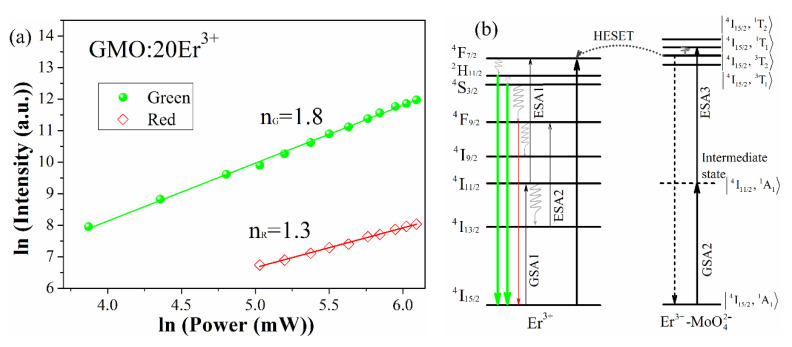
Pump power dependence of the green and red emission intensities in Gd_2_(MoO_4_)_3_:20Er^3+^ phosphor (**a**). Energy-level diagrams and possible upconversion mechanism for Gd_2_(MoO_4_)_3_:Er^3+^ phosphor (**b**).

**Figure 5 nanomaterials-12-03641-f005:**
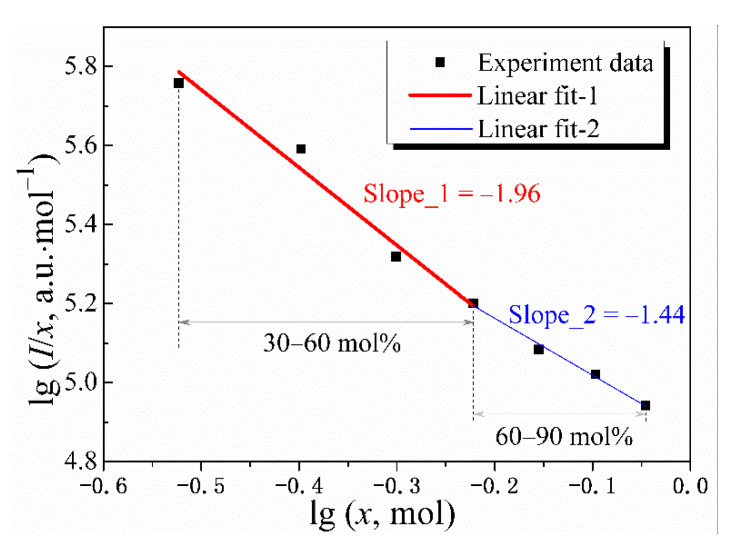
Plots of lg(*I*/*x*) versus lg*x* of Gd_2_(MoO_4_)_3_:*x*Er^3+^ phosphors under 976 nm excitation.

## Data Availability

Not applicable.

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
