# Peer review of "Optical Properties and Concentration Quenching Mechanism of Er3+ Heavy Doped Gd2(MoO4)3 Phosphor for Green Light-Emitting Diode"

_nanomaterials, 2022, doi:10.3390/nano12203641_

Round 1

Reviewer 1 Report

In this work authors studied the luminesce properties, phase composition and morphology of microcrystalline Gd2(MoO4)3:Er3+  upconverting phosphors synthesized  by sol-gel method. The materials are carefully characterized by luminescence spectroscopy, XRD, SEM, EDX methods, and the observed results are explained. Authors found that Er3+ concentration up to 20 mol% did not cause concentration quenching. The upconversion and concentration quenching mechanisms were proposed by the authors. The article is well written and interesting for broad range of readers. The article is probably publishable, but it has some issues that should be resolved:

1)           Introduction: Authors should perform the literature search and summarize the previous study on Gd2(MoO4)3:Er3+  materials citing the corresponding articles. For example, 10.1007/s10895-008-0414-2 and 10.1007/10857522_29. Also, authors must explain the difference between their results reported in the current paper and previously published works. Thus, for example, in work of X. F. Liang et al. (10.1007/s10895-008-0414-2) about Gd2(MoO4)3:Er3+ materials, the maximum of upconversion emission intensity is reached at 20-30% of Er3+ similarly to the current work.

2)           Section 3.1. Crystal Structure and Morphology: please provide the average particle size.

3)           Section 3.2. Upconversion Mechanism Analysis: please discuss in details the upconversion mechanism provided in Fig. 4b taking special attention how the obtained data supports the provided mechanism.

4)           Line 88: Typo. Should be concentration instead “concentra tion”.

Author Response

Reviewer 1

Comments and Suggestions for Authors

In this work authors studied the luminesce properties, phase composition and morphology of microcrystalline Gd2(MoO4)3:Er3+ upconverting phosphors synthesized  by sol-gel method. The materials are carefully characterized by luminescence spectroscopy, XRD, SEM, EDX methods, and the observed results are explained. Authors found that Er3+ concentration up to 20 mol% did not cause concentration quenching. The upconversion and concentration quenching mechanisms were proposed by the authors. The article is well written and interesting for broad range of readers. The article is probably publishable, but it has some issues that should be resolved:

1)  Introduction: Authors should perform the literature search and summarize the previous study on Gd2(MoO4)3:Er3+ materials citing the corresponding articles. For example, 10.1007/s10895-008-0414-2 and 10.1007/10857522_29. Also, authors must explain the difference between their results reported in the current paper and previously published works. Thus, for example, in work of X. F. Liang et al. (10.1007/s10895-008-0414-2) about Gd2(MoO4)3:Er3+ materials, the maximum of upconversion emission intensity is reached at 20-30% of Er3+ similarly to the current work.

Response 1: Thanks for the comment. We read the recommended works carefully that are helpful for this manuscript and cited them as references. And the following passage has been added to the manuscript. “Gd2(MoO4)3 is an excellent host due to its relatively small maximum phonon energy and good thermal stability. In addition, Gd3+ ions and activated ions (RE3+) are both rare earth ions and have similar chemical properties. Therefore, it can achieve high concentration doping in Gd2(MoO4)3 host, which increases the number of luminescent centers and enhances the upconversion luminescence intensity. So far, there are few reports on the upconversion luminescence of Er3+ single-doped Gd2(MoO4)3 [17]. The upconversion luminescence properties of Gd2(MoO4)3:Er3+ phosphor under excitation at 1510-1565 nm had been analyzed [17].

However, for the most widely used ~980nm excitation wavelength, it is necessary to study the upconversion luminescence characteristics and concentration quenching analysis of Er3+ heavy doped Gd2(MoO4)3 phosphors.” (line 44-53)

2)  Section 3.1. Crystal Structure and Morphology: please provide the average particle size.

Response 2: Thanks for the comment. The sentence “and the average particle size is about 0.9 μm (see inset of Figure 1(b)). ” has been added to the manuscript (line 86).

Figure 1. XRD patterns of Gd2(MoO4)3:xEr3+ (x = 0, 10, 20, 40, 60, 80, 90 mol%) phosphors (a). SEM image (b) and EDS spectrum (c) of Gd2(MoO4)3:20Er3+ phosphor. The inset of (b) is particle size distribution.

3)  Section 3.2. Upconversion Mechanism Analysis: please discuss in details the upconversion mechanism provided in Fig. 4b taking special attention how the obtained data supports the provided mechanism.

Response 3: Thanks for this good suggestion. The details on the upconversion mechanism provided in Fig. 4b have been added to the manuscript. “The 2H11/2/4S3/2 and 4F9/2 levels of Er3+ ions are populated via multiphonon relaxation from the 4F7/2 level, and the green and red upconversion emissions are observed. As shown in Figure 4(b), two possible upconversion passways populate the 4F7/2 level of Er3+. One is that the ground state 4I15/2 excites to the 4I11/2 level via ground state absorption (GSA1), and then further excites to the 4F7/2 level via excited state absorption (ESA1). The other is supposed to be a high excited state energy transfer (HESET). The Er3+–MoO42− dimer complex absorbs an infrared photon and is excited from the ground state |4I15/2, 1A1> to the intermediate state |4I11/2, 1A1> via GSA2, under a 980 nm excitation. Subsequently, the |4I15/2, 1T1> state is further excited from the |4I11/2, 1A1> state via ESA3, and then decays nonradiative to the |4I15/2, 3T2> state. The energy transfer from |4I15/2, 3T2> to the 4F7/2 level of Er3+ vis the HESET process is efficient. Thus, the populated 4F7/2 level relax nonradiative by a fast multiphonon decay process to the 2H11/2/4S3/2 and 4F9/2 levels. And then the electrons transition from the excited state (2H11/2/4S3/2 and 4F9/2) to the ground state, accompanied by intense green and weak red emissions, in agreement with our experimental results (see Figure 4(a)).” (line 140-154)

4)  Line 88: Typo. Should be concentration instead “concentra tion”.

Response 4: We apologize for this error and carefully correct this mistake in the manuscript.

Reviewer 2 Report

The manuscript by Li et al describes the up-conversion properties of Er3+ doped Gd2(MoO4)3 molybdate for the purpose of green LED.

This is an original and interesting work since only few articles are published concerning simply doped Gd2(MoO4)3 : Er3+ phosphors. The aim of this work is thus clearly exposed in introduction.

However, to improve the quality of the paper I recommend a major revision for the following reasons:

- Gd2(MoO4)3 exhibits a monoclinic structure with actually a C2/c space group. The orthorhombic phase is not observed at room temperature. Please correct your text (lines 15, 42, 65, 188). beta angle is missing (lines 66 and 173). Please give a reference for the structural data.

- Introduction: refs 13 and 14 refer to Nd3+ ion, not to Yb3+/RE3+. Please correct.

- Strictly speaking x must be considered as a doping rate (see equations 3 and 4). I think x values in Fig. 5 are wrong. For instance, for a 30 mol% doping, the x value should be considered as 0.6 and not 0.3 due to the host formula Gd2(MoO4)3. Please check your results and recalculate Rc value. Modify your comments in lines 174-177 if necessary.

- There are no comments on Fig. 3b (CIE diagram).

- Fig. 4a: the 1.3 value for the red emission is unusually low. Please check your results or give some explanation.

- Comments on Fig. 4b should be strengthened. Please describe the UC process in a few lines especially concerning the Er-MoO4 energy level diagram. What is HESET?

Author Response

Reviewer 2

Comments and Suggestions for Authors

The manuscript by Li et al describes the up-conversion properties of Er3+ doped Gd2(MoO4)3 molybdate for the purpose of green LED.

This is an original and interesting work since only few articles are published concerning simply doped Gd2(MoO4)3 : Er3+ phosphors. The aim of this work is thus clearly exposed in introduction.

However, to improve the quality of the paper I recommend a major revision for the following reasons:

1)  Gd2(MoO4)3 exhibits a monoclinic structure with actually a C2/c space group. The orthorhombic phase is not observed at room temperature. Please correct your text (lines 15, 42, 65, 188). beta angle is missing (lines 66 and 173). Please give a reference for the structural data.

Response 1: We are sorry for our unclear expression. The corresponding contents have been revised in the manuscript. Meanwhile, the XRD patterns in our work were very similar to the reported work (Opt. Express, https://doi.org/10.1364/OE.24.0A1276), confirming the high purity of as-prepared monoclinic samples.

2)  Introduction: refs 13 and 14 refer to Nd3+ ion, not to Yb3+/RE3+. Please correct.

Response 2: Thanks for the comment. The papers [10,13-16] refer to Yb3+/RE3+ heavy doped, and they have been cited in the manuscript. (Line 40)

[10]  Wen, S.; Zhou, J.; Zheng, K.; Bednarkiewicz, A.; Liu, X.; Jin, D. Advances in highly doped upconversion nanoparticles. Nature communications 2018, 9, 1-12.

[13]  Liu, L.; Lu, K.; Xu, L.; Tang, D.; Liu, C.; Shahzad, M.K.; Yan, D.; Khan, F.; Zhao, E.; Li, H. Highly efficient upconversion luminescence of Er heavily doped nanocrystals through 1530 nm excitation. Optics Letters 2019, 44, 711-714.

[14]  Wen, S.; Li, D.; Liu, Y.; Chen, C.; Wang, F.; Zhou, J.; Bao, G.; Zhang, L.; Jin, D. Power-Dependent Optimal Concentrations of Tm3+ and Yb3+ in Upconversion Nanoparticles. The Journal of Physical Chemistry Letters 2022, 13, 5316-5323.

[15]  Johnson, N.J.; He, S.; Diao, S.; Chan, E.M.; Dai, H.; Almutairi, A. Direct evidence for coupled surface and concentration quenching dynamics in lanthanide-doped nanocrystals. Journal of the American Chemical Society 2017, 139, 3275-3282.

[16]  Liu, Y.; Pan, G.; Gao, H.; Zhang, H.; Ao, T.; Gao, W.; Mao, Y. Single band red emission of Er3+ ions heavily doped upconversion nanoparticles realized by active-core/active-shell structure. Ceramics International 2021, 47, 18824-18830.

3) Strictly speaking x must be considered as a doping rate (see equations 3 and 4). I think x values in Fig. 5 are wrong. For instance, for a 30 mol% doping, the x value should be considered as 0.6 and not 0.3 due to the host formula Gd2(MoO4)3. Please check your results and recalculate Rc value. Modify your comments in lines 174-177 if necessary.

Response 3: Thanks for the comment. The corresponding contents have been revised in the manuscript.

4)  There are no comments on Fig. 3b (CIE diagram).

Response 4: A description corresponding to Fig. 3b has been added. “The calculated chromaticity coordinate of Gd2(MoO4)3:20Er3+ phosphor is X = 0.23 and Y = 0.74, which fall exactly in the green region of CIE chromaticity diagram (see Figure 3(b)). ” (Line 128-130)

5)  Fig. 4a: the 1.3 value for the red emission is unusually low. Please check your results or give some explanation.

Response 5: Thanks for the comment. The 1.3 value for the red emission is unusually low, which may be caused by high excited state energy transfer (HESET) (doi.org/10.1016/j.jallcom.2015.07.005) and the saturation of the energy upconversion process (doi:10.1103/PhysRevB.61.3337). Due to the effective energy transfer between a sensitizer (Er3+–MoO42 dimer) and an activator (Er3+), HESET from the |4I15/2, 3T2> state of Er3+–MoO42− dimer to the 4F7/2 level of Er3+ ion is dominant. The lower emitting levels (2H11/2/4S3/2 and 4F9/2 levels of Er3+) are then populated via multiphonon relaxation. And the electrons transition from the excited state (2H11/2/4S3/2 and 4F9/2) to the ground state, accompanied by intense green and weak red emissions. Because HESET is extremely efficient, the saturation of the energy upconversion process takes place in the case of relatively low excitation power. The value of nR deviates greatly from 2 in the graph of the ln–ln relationship, attributing to the competition between the linear decay and the upconversion process for the depletion of the intermediate excited states.

6)  Comments on Fig. 4b should be strengthened. Please describe the UC process in a few lines especially concerning the Er-MoO4 energy level diagram. What is HESET?

Response 6: Thanks for this good suggestion. HESET is high excited state energy transfer. The details on the upconversion mechanism provided in Figure 4(b) have been added to the manuscript.The 2H11/2/4S3/2 and 4F9/2 levels of Er3+ ions are populated via multiphonon relaxation from the 4F7/2 level, and the green and red upconversion emissions are observed. Two possible upconversion passways for populating the 4F7/2 level of Er3+. One is that the ground state 4I15/2 excites to the 4I11/2 level via ground state absorption (GSA1), and then further excites to the 4F7/2 level via excited state absorption (ESA1). The other is supposed to be a high excited state energy transfer (HESET). The Er3+–MoO42− dimer complex absorbs an infrared photon and is excited from the ground state |4I15/2, 1A1> to the intermediate state |4I11/2, 1A1> via GSA2, under a 980 nm excitation. Subsequently, the |4I15/2, 1T1> state is further excited from the |4I11/2, 1A1> state via ESA3, and then decays nonradiative to the |4I15/2, 3T2> state. The energy transfer from |4I15/2, 3T2> to the 4F7/2 level of Er3+ vis the HESET process is efficient. Thus, the populated 4F7/2 level relax nonradiative by a fast multiphonon decay process to the 2H11/2/4S3/2 and 4F9/2 levels. And then the electrons transition from the excited state (2H11/2/4S3/2 and 4F9/2) to the ground state, accompanied by intense green and weak red emissions, in agreement with our experimental results (see Figure 2(a)).” (line 141-155)

Reviewer 3 Report

My comments are in attached file.

Author Response

Reviewer 3

This manuscript contains a short description of the conditions used for the preparation of Gd2(MoO4)3:Er3+ phosphors by the sol-gel synthesis and high-temperature treatment. Structural characteristics of the products were obtained by the X-ray diffraction analysis. Micromorphology and element composition were observed by SEM/EDS methods. Optical frequency upconversion properties of the phosphors excited at 976 nm were determined by conventional experimental methods. The experimental conditions used for the sample preparation and property measurements are given in detail that opens a possibility for future testing of the results by other researchers. In my opinion, this study is interesting because wide-range rare-earth element substitution was reached in the phosphor family and the related effects were evaluated. The general level of this study is high and manuscript could be considered for publication after minor revision reasonable to increase the paper quality. My several local corrections proposed for the text and questions are listed below for author consideration.

  • “ In this paper, Er3+ heavy doped in Gd2(MoO4)3 phosphor with orthorhombic phase were prepared by a sol-gel process.  

In this study, Er3+ heavy doped in Gd2(MoO4)3 phosphor with orthorhombic phase were prepared by a sol-gel process.  ”

  1.   “Gd2(MoO4)3:Er3+ phosphor emitted remarka-16 bly intense green emission under excitation at 976 nm, and Er3+ concentration up to 20 mol% did not cause concentration quenching.

Gd2(MoO4)3:Er3+ phosphor emitted remarka-16 bly intense green emission under the excitation at 976 nm, and Er3+ concentration up to 20 mol% did not cause concentration quenching. ”

  1. “The Er3+ concentration was in the range of 30-60 mol%, the concentration quenching belonged to the electric dipole-dipole interaction, and the concentration greater than 60 mol% belonged to the exchange interaction.

When the Er3+ concentration was in the range of 30-60 mol%, the concentration quenching was governed by the electric dipole-dipole interactions, and, when the concentration was greater than 60 mol%, the concentration quenching was controlled by the exchange interactions.”

  1. “Lanthanide-doped upconversion materials capable of converting low-energy near-infrared light to high-energy visible light, are potential candidates for white light display[1], the upconversion laser[2], solar cell[3], anti-counterfeit labels[4], Photocatalytic[5], volumetric color imaging[6], temperature easurement[7], super-resolution imaging[8], and biological fluorescence labeling[9]. However, some problems limit their applications, such as the low upconversion emission intensity.

Lanthanide-doped upconversion materials capable of converting low-energy near-infrared light to high-energy visible light, are potential candidates for white light displays[1], upconversion lasers[2], solar cells[3], anti-counterfeit labels[4], photocatalytic structures[5], volumetric color imaging[6], temperature measurement[7], super-resolution imaging[8] and biological fluorescence labeling[9]. However, some problems limit their applications, including the low upconversion emission intensity.”

  1. “In this work, we prepared heavy Er3+ doped in Gd2(MoO4)3 phosphor with orthorhombic structure using the sol–gel method.

In this work, we prepared heavy Er3+-doped Gd2(MoO4)3-based phosphor with orthorhombic structure using the sol–gel method.”

  1. “And the concentration quenching mechanism was analyzed.

Also, the concentration quenching mechanism was analyzed.”

  1. “The Gd2(MoO4)3:x mol% Er3+ (x=1, 5, 10, 15, 20, 25, 30, 35, 40, 50, 60, 70, 80, 90 mol%) phosphors were prepared using a simple sol–gel process.

The Gd2(MoO4)3:x Er3+ (x=1, 5, 10, 15, 20, 25, 30, 35, 40, 50, 60, 70, 80, 90 mol%) phosphors were prepared using a simple sol–gel process.”

Response 1: Thanks for these good suggestions. The corresponding contents have been revised in the manuscript.

2) a. According to our previous 48 work[15], Gd(NO3)3·6H2O (AR), (NH4)6Mo7O24 (AR), and Er(NO3)3·6H2O (AR) with precalculated concentrations were dissolved in deionized water and the citric acid (AR) was added to this solution. Besides purity, supplier should be reported for each starting reagent.

Response 2: Thanks for the comment. The sentence “All raw materials were purchased from Sinopharm.” has been added to the manuscript.

3) a. “Subsequently, the pH of the solution was adjusted to about 7 by adding an appropriate amount of ammonia (AR).

Subsequently, pH level of the solution was adjusted to about 7 by adding an appropriate amount of ammonia (AR).”

  1. “The scanning electron microscope (SEM) (JSM-5510; JEOL, Japan) equipped with an energy 58 dispersive X-ray spectrum (EDS) was used to characterize the microstructures.

The scanning electron microscope (SEM) (JSM-5510; JEOL, Japan) equipped with an energy 58 dispersive X-ray spectrum (EDS) analyzer was used to characterize the microstructures.”

Response 3: Thanks for these good suggestions. The corresponding contents have been revised in the manuscript.

4). The XRD patterns of Gd2(MoO4)3:xEr3+ phosphors are shown in Figure 1(a). The XRD patterns are well matched with the standard data (JCPDS No. 26-0655), indicating that a single orthorhombic phase Gd2(MoO4)3:Er3+ phosphor have been acquired.

XRD patterns of Gd2(MoO4)3:xEr3+ phosphors are shown in Figure 1(a). The XRD patterns are well matched with the standard data (JCPDS No. 26-0655), indicating that a single orthorhombic phase Gd2(MoO4)3:Er3+ phosphor is acquired.

Besides card number, the related original paper should be cited.

Response 4: We are sorry for our unclear expression. The corresponding contents have been revised in the manuscript. Meanwhile, the XRD patterns in our work were very similar to the reported work (Opt. Express, https://doi.org/10.1364/OE.24.0A1276) that was cited as a reference, confirming the high purity of as-prepared monoclinic samples.

5) These phosphors have a space group of C2/c, with the cell parameters of a = 7.53, b = 11.38, c = 11.40.

These phosphors have a space group of C2/c, with the cell parameters of a = 7.53, b = 11.38 and c = 11.40 A.

Is it right that the cell parameters are the same in all samples, independently of Er content? As it was shown in many studies, cell parameters are commonly affected by rare-earth substitution:

  1. Solid State Chem. 228 (2015) 160-166

Molecules 26 (2021) 7357.

Response 5: Thanks for the comment. The substitution of big Gd3+ ion by smaller Er3+ ion is reasonably resulted in the unit cell shrinkage. The sentence “The monoclinic Gd2(MoO4)3 have a space group of C2/c, with the cell parameters of a = 7.53, b = 11.38, c = 11.40 Å.” has been added to the manuscript.

6) No other impurity peaks are observed, indicating that the doped Er3+ ions have been incorporated into the lattice, replacing the lattice position of Gd3+.

What is the confirmation that Er3+ ions are substituted for Gd3+ ions? What is the relation between nominal composition of the reaction solution and Er/Gd ratio in the doped samples?

Response 6: Thanks for the comment. It is worth mentioning that trivalent Gd and Er ions have the similar electronic structures (s2p6), as well as their effective ionic radius (93.8 pm for Gd3+, 89.0 pm for Er3+). Typically, Er3+ ions could enter into Gd2(MoO4)3 lattice by substituting Gd3+ sites. The relation between nominal composition of the reaction solution and Er/Gd ratio in the doped samples is in accordance with the stoichiometric ratio.

7). a. “The morphology and the elemental components of Gd2(MoO4)3:20Er3+ phosphor were identified by SEM image and EDS spectrum , as shown in Figure 1(b) and (c).

The morphology and the elemental components of Gd2(MoO4)3:20Er3+ phosphor were identified by SEM and EDS measurements , as shown in Figure 1(b) and (c).”

  1. “To reveal the concentration-dependent upconversion luminescence properties, upconversion luminescence spectra in Gd2(MoO4)3:xEr3+ phosphors under a 976 nm LD excitation are shown in Figure 2(a).

To reveal the concentration-dependent upconversion luminescence properties, the upconversion luminescence spectra in Gd2(MoO4)3:xEr3+ phosphors under the 976 nm LD excitation are shown in Figure 2(a).”

  1. “The green emission intensity is quite strong relative to the red emission intensity, as shown in Figure 2(b).

The green emission intensity is quite strong in reference to the red emission intensity, as shown in Figure 2(b).”

  1. “When the Er3+ concentration increases to 20 mol%, the green and red upconversion emission intensities reach the strongest, and then decreased with increasing concentra tion.

When Er3+ concentration increases to 20 mol%, the green and red upconversion emission intensities are the strongest, and then they decrease with increasing Er concentration.”

  1. “Interestingly, the red emission intensity increases instead.

Interestingly, the red emission intensity increases in this concentration range.”

  1. “This may be because the chemical properties of Er3+ ions and Gd3+ ions are similar, and the radius is almost the same, so Er3+ ions can easily replace Gd3+ ions in the matrix and enter the lattice.

This may be because the chemical properties of Er3+ ions and Gd3+ ions are similar, and the radii are almost the same. So, Er3+ ions can easily replace Gd3+ ions in the matrix and enter the lattice.”

  1. “The intensity of green emission (left) and red emission (light) as a function of Er3+ concentrations in Gd2(MoO4)3:xEr3+ phosphors (c).

The intensity of green emission (left) and red emission (right) as a function of Er3+ concentration in Gd2(MoO4)3:xEr3+ phosphors (c).”

  1. “With the increase of Er3+ concentration from 1 mol% to 20 mol%, the GRR correspondingly increased from 33.2 to 105.2.

With the increase of Er3+ concentration from 1 to 20 mol%, the GRR is increased from 33.2 to 105.2.”

  1. “Continue to increase the Er3+ concentration, and GRR shows a decreasing trend.

If to continue increase of the Er3+ concentration, GRR shows a decreasing trend.”

  1. “where I is the upconversion emission intensity, P is the pump laser power, the n is the number of pumping photons required in the upconversion mechanism.

where I is the upconversion emission intensity, P is the pump laser power, the n is the number of pumping photons required in the upconversion process.”

  1. “The possible upconversion mechanism for Gd2(MoO4)3:Er3+ phosphor is shown in Figure 4(b), and the detailed upconversion mechanism is described in the works of literature [17] and [18].

The possible upconversion mechanism for Gd2(MoO4)3:Er3+ phosphor is shown in Figure 4(b), and the detailed upconversion mechanism is described in the literature [17,18].”

  1. “According to Dexter's theory, the functional relationship between the fluorescence intensity I and the activator concentration x can be expressed as[16]

According to the Dexter theory, the functional relationship between the fluorescence intensity I and the activator concentration x can be expressed as[16]”

  1. “Based on the upconversion emission spectra, plots of lg(I/x) versus lgx were depicted in Figure 5.

Based on the upconversion emission spectra, plots of lg(I/x) versus lgx are depicted in Figure 5.”

  1. “Therefore, in Gd2(MoO4)3:Er3+ phosphor, the fluorescence quenching caused by Er3+ concentration of 30-60 mol% belonged to the electric dipole-dipole interaction.

Therefore, in Gd2(MoO4)3:Er3+ phosphor, the fluorescence quenching caused by Er3+ concentration in the range of 30-60 mol% is governed by the electric dipole-dipole interaction.”

  1. “When the Er3+ concentration increased from 60 mol% to 90 mol%, the corresponding s value was 4.32 (1.44×3), which approached the theoretical value of 3 for the exchange interactions. Therefore, it was inferred that the fluorescence quenching was mainly caused by the exchange interaction in this concentration range.

When Er3+ concentration is increased from 60 to 90 mol%, the corresponding s value is 4.32 (1.44×3), which approaches the theoretical value of 3 of the exchange interactions. Therefore, it is inferred that, in this concentration range, the fluorescence quenching is mainly caused by the exchange interaction.”

  1. “Rc has the following formula:[19]

Rc can be calculated by the following formula:[19]”

  1. “Figure 5. Plots of lg(I/x) versus lgx of Gd2(MoO4)3:xEr3+ phosphors under 976 nm excitation.

Figure 5. Plots of lg(I/x) versus lgx of Gd2(MoO4)3:xEr3+ phosphors under the 976 nm excitation.”

  1. “where V is the volume of the unit cell, x is the activator ions concentration and N refers to the number of cations in the unit cell.

where V is the unit cell volume, x is the activator ions concentration and N refers to the number of cations in the unit cell.”

  1. “Considering that the radius of Er3+ ions is 0.89 Å, which is slightly smaller than the Gd3+ ion radius of 0.94 Å.

The radius of Er3+ ions is 0.89 Å, which is slightly smaller than the Gd3+ ion radius of 0.94 Å.”

  1. “ And the Er3+ concentration is as high as 60 mol%, it is reasonable to believe that the calculated Rc is larger than the theoretical value.

When the Er3+ concentration is as high as 60 mol%, it is reasonable to believe that real Rc value is larger than the theoretical value.”

  1. “In summary, Gd2(MoO4)3:xEr3+ phosphors with orthorhombic phase have been prepared by a simple sol-gel method.

In summary, Gd2(MoO4)3:xEr3+ phosphors with orthorhombic phase were prepared by the simple sol-gel method.”

  1. “The phosphors exhibit intense upconversion green emission under 976 nm excitation.

The phosphors exhibit intense upconversion green emission under the 976 nm excitation.”

  1. “The Er3+ concentration is in the range of 30-60 mol%, the concentration quenching progress is mainly due to the electric dipole-dipole interaction, and the Er3+ concentration greater than 60 mol% is due to the exchange interaction.

When the Er3+ concentration is in the range of 30-60 mol%, the concentration quenching progress is governed by the electric dipole-dipole interaction, and, when the Er3+ concentration is greater than 60 mol%, due to the exchange interaction.”

  1. “The results show that Gd2(MoO4)3:Er3+ powder is an excellent green phosphor, and Gd2(MoO4)3 material as an excellent host for lanthanide heavy doping has great potential in solid-state lighting.

The results show that Gd2(MoO4)3:Er3+ powder is an excellent green phosphor, and Gd2(MoO4)3 material as an excellent host for lanthanide heavy doping.”

  1. “This research was supported by National Natural Science Foundation of China 203 (61705095, 12004093);

This research was supported by the National Natural Science Foundation of China 203 (61705095, 12004093);”

Response 7: Thank you very much for your careful review. We have revised the whole manuscript carefully and corrected some grammar errors and instances of badly worded/constructed sentences.

8) The prepared phosphor is composed of tightly packed particles with relatively uniform morphology and micro-size.

It could be emphasized that such morphology is commonly observed in molybdate and tungstate phosphors synthesized by the sol-gel method with the following annealing:

Phys. Chem. Chem. Phys. 17 (2015) 19278-19287

Mater. Lett. 181 (2016) 38-41

Korean J. Mater. Res. 29 (12) (2019) 741-746

Response 8: Thanks for the good suggestion. We read these documents carefully, found them helpful for this manuscript improvement, and included them as the references.

Round 2

Reviewer 1 Report

The manuscript was carefully revised and currently can be accepted in the present form.

Author Response

Thank you very much.

Reviewer 2 Report

The work of Li et al can be accepted for publication taking into account the following minor corrections:

- Gd2(MoO4)3 crystal data : beta angle is still missing (lines 79 and 204 in revised text).

- Fig. 4b comments: the ESA2 mechanism has been forgotten.

Author Response

The work of Li et al can be accepted for publication taking into account the following minor corrections:

1)  Gd2(MoO4)crystal data : beta angle is still missing (lines 79 and 204 in revised text).

Response 1: Thanks for the comment. The corresponding contents have been revised in the manuscript. “The monoclinic Gd2(MoO4)3 have a space group of C2/c, with the cell parameters of a = 7.53, b = 11.38, c = 11.40 Å, β = 109.32 degree.”   “(a = 7.55, b = 11.44, c = 11.47 Å, V = 934.51 Å3, β = 109.32 degree, Z = 4)”

2) Fig. 4b comments: the ESA2 mechanism has been forgotten.

Response 2: Thanks for this good suggestion. The sentence “Besides, the 4F9/2 (Er3+) is populated via ESA2 that involved the 4F9/2 4I13/2 transition.” has been added to the manuscript.